# Personalized Diet in Obesity: A Quasi-Experimental Study on Fat Mass and Fat-Free Mass Changes

**DOI:** 10.3390/healthcare9091101

**Published:** 2021-08-25

**Authors:** Manuel Reig García-Galbis, Diego I. Gallardo, Rosa María Martínez-Espinosa, María José Soto-Méndez

**Affiliations:** 1Department of Nutrition and Dietetics, Faculty of Health Sciences, Isabel I University, 09003 Burgos, Spain; 2Department of Mathematics, University of Atacama, Copiapó 1530000, Chile; diego.gallardo@uda.cl; 3Faculty of Medicine, University of Atacama, Copiapó 1530000, Chile; 4Department of Agrochemistry and Biochemistry, Division of Biochemistry and Molecular Biology, Faculty of Sciences, University of Alicante, 03690 Alicante, Spain; rosa.martinez@ua.es; 5Iberoamerican Nutrition Foundation (FINUT), 18016 Granada, Spain; msoto@finut.org

**Keywords:** obesity, overweight, management, clinical practice guidelines, fat mass, adiposity, fat-free mass, sarcopenia, Spain

## Abstract

Considering that the prevalence of overweight and obesity in Southeast of Spain is high, the aim of this work was to analyze the relation between the adherence to a personalized diet and the effectiveness of changes in the body composition in overweight and obese adults in this region. This quasi-experimental study presents the following selection criteria: attendance at the consultation between 2006 and 2012, subjects ≥ 19 years of age with overweight or obesity. In total, 591 overweight or obese individuals were involved in this study, attending 4091 clinic consultations in total. Most of the sample consisted of subjects who attended >3 consultations (>1.5 months), and were females aged 19–64 years who obtained clinically significant changes in fat mass (FM, ≥5%) and recommended changes in fat-free mass (FFM, ≥0%). Based on the results obtained and the experience gained from this research, the following recommendations are established: (i) record fat mass and fat-free mass index as a complement to body mass index; (ii) use FM and FFM to evaluate effectiveness of interventions with the aim of obtaining body composition changes; (iii) use personalized diet to achieve significant changes in FM and avoid non-recommended changes in FFM.

## 1. Introduction

Obesity is a condition in which there is an excessive accumulation of fat that can have negative effects on health and decrease life expectancy. The growth of adipose tissue increases the risk of developing age-related pathologies (cardiovascular diseases, type 2 diabetes, musculoskeletal disorders, respiratory diseases, and various types of cancer) [1]; these factors are exacerbated when mixed with aging [2]. In 2014, 375 million women and 266 million men were obese worldwide, compared to 71 million women and 34 million men in 1975 [3]. In 2017, the Spanish National Health Survey (ENSE) confirmed that in the last 30 years the prevalence of obesity in adults had increased from 7.4% to 17.4% (comparison between 1987 and 2017) [4].

The prevalence of obesity and overweight is higher in men than in women, especially affecting lower income classes [4,5] (p. 58). A recent review in Spain identified an economic cost overrun in the medical care received by overweight adults of €1.95 billion/year in 2016; if the current trend of increasing prevalence of overweight individuals continues, it will represent a cost overrun of €3 billion/year by 2030 [6]. Between 2014 and 2015, the Spanish Agency for Consumer Affairs, Food Safety and Nutrition (AECOSAN) developed the ENALIA 2 project, in which a national food survey was conducted, including subjects aged 18 to 75 years. The results obtained showed that there is a deviation in the recommended calorie profile, since the intake of protein and fat is higher than that of carbohydrates. Based on these results, it was recommended to increase their consumption through whole grains, fruits, and vegetables [7].

To cope with overweight and obesity, the most effective options are interventions that include lifestyle modification, diet, and exercise planning [8]. A personalized diet helps to improve adherence to the recommendations through consultation and avoid body weight regain, therefore gaining greater long-term benefits [9]. Behavioral support is a catalyst in incorporating dietary recommendations into day-to-day living, and will help remove barriers in lifestyle change [8,10,11]. According to different studies, the main barriers found in young adults have been a lack of male interest towards diet; an unhealthy diet; lack of time to plan the grocery shopping, preparation, and cooking of healthy food; lack of facilities to store, prepare, and cook healthy food; or limited knowledge, skills, and motivation for the incorporation of healthy habits. The key factors for an improvement in diet monitoring were as follows: female interest in a healthy diet; support and encouragement from friends and family to eat healthily; desire to improve health, control body weight, and improve self-esteem to be accepted among potential partners and others; possessing a motivation to eat healthily and existence of self-regulation skills [12].

Body composition monitoring is an important part of the follow-up in an intervention in overweight subjects [13]. Scheduling realistic expectations in body weight changes, as well as changes occurring throughout the intervention are determining factors to improve adherence [14,15]. Sometimes, achieving an ideal body weight is difficult [16,17]; consequently, it is recommended to obtain changes between 5 and 10% of body weight, as this is associated with significant improvements for health [8,16,18,19]. These body weight changes have been defined as clinically significant, however, clinically significant changes in fat mass are still an underexplored area [8,16,17,18,19,20,21,22,23]. Because of this lack of details on clinically significant changes in fat mass, it is important to justify the reasons explaining why changes in fat mass are more appropriate and accurate than body weight changes. The main explanations for this justification are: (i) body weight changes could be occurring due to non-recommendable changes in fat-free mass [24,25]; (ii) changes in fat mass have been associated with the improvement of metabolic alterations and with the development of certain pathologies [1,2,13,26,27]; (iii) the recommendation of the American Association of Clinical Endocrinologists and American College of Endocrinology of introducing fat mass analysis as part of the standardization of intervention protocols for overweight and obesity [28] (p. 376); (iv) the maintenance of a cut-off point of ≥5% as a clinically significant change was considered an adequate technique to evaluate cases of overweight or overfat. In the analysis of changes in fat-free mass, little information is observed, identifying that its decrease contributes to the prediction of metabolic syndrome [29,30]; therefore, any decrease should be considered as an undesirable change. In the analysis of fat mass, its index (fat mass (kg) divided by height (m^2^)) shows greater accuracy in the diagnosis of metabolic disorders than other variables, such as BMI, waist-to-hip ratio, and fat mass [31,32]. In the analysis of fat-free mass, its index (fat-free mass (kg) divided by height (m^2^)) is a variable that allows a more precise assessment of lean mass depletion [30] (p. 381). Therefore, fat and fat-free mass indexes are body composition variables that should be used in interventions with overweight subjects.

Considering all the previous aspects, this study proposes two hypotheses: (i) personalized diet is the best catalyst for changes in body composition; (ii) subjects who participate will get greater fat mass changes, but minor changes in fat-free mass. Both hypotheses would help to counteract the high prevalence of overweight and obesity. The main objective of the study was to analyze the relationship between adherence to a personalized diet and the effectiveness of changes in body composition in overweight and obese adults in the Southeast of Spain (Alicante). Secondary objectives: (i) to record differences between body composition parameters; (ii) to analyze the differences between body composition changes produced because of attendance to the consultation; (iii) to identify the characteristics of the subjects reaching non-recommendable changes in fat-free mass.

## 2. Materials and Methods

### 2.1. Study Design

The research carried out was defined as a quasi-experimental study. The consultation center was in the south-east of Spain (Alicante), and data were recorded between the years 2006–2012. The ethics committee of the University of Alicante approved this study (UA−21 April 2006) and the recommendations of the Declaration of Helsinki and the International Committee of Biomedical Journal Editors were followed [33,34]. The selection criteria related to the subjects involved in the study were: (i) attendance to the dietitian consultation; (ii) subjects aged 19–86 years (19–64 years vs. ≥65 years); (iii) subjects with overweight and obesity [35,36] (p. 184) (Table 1). An adult was defined as overweight when the BMI ranged between 25 and 29.9, and obese when ≥30 [36] (p. 184). Exclusion criteria: underweight and healthy weight subjects; subjects with food allergies or intolerances, physical, or mental disability or any medical limitation to perform this type of intervention; subjects who did not sign the informed consent.

This methodology is based on a dynamic learning process, in which three periods were differentiated: initiation, improvement, and maintenance. Maintenance or progression from one period to the next depended on the changes in body weight and fat mass produced, which were related to the incorporation of recommended dietary changes in their daily lives. (Appendix A) [8,13,37,38,39].

### 2.2. First Consultation

It was advisable to attend at least 2 consultations per month (30 min/consultation). At this time, the following data were monitored: (i) history of diseases; (ii) food history (tastes, habits, and food preferences); (iii) food and liquids consumed according to the daily schedule; (iv) person usually cooking at home, number of meals consumed per day, and number of meals consumed at home or outside; (v) poor digestion or constipation problems; whether physical activity is performed; (vi) hours of sleep (sleeping all night or waking up often) and oral health problems; and (vii) biochemical markers from a recent blood test (Appendix A) [8,12,13,37,38,39]. Based on performance protocols, the following parameters were measured: (i) height (height rod Seca 206); (ii) body weight, as part of the body composition (bioimpedance device BC−418 MA (Tanita Corporation of America)); and (iii) blood pressure in the right arm, seated and at the height of the heart (Littmann fonendoscope and Riester bracelet) [40].

Bioimpedance was selected for body composition measurement because it is a fast method with acceptable accuracy [40,41,42] (p. 183). Here, accuracy is defined as the variation of body composition between different measurements in the same subject [41]. The impedance measurement protocol was [43]: (i) age and sex were recorded; (ii) height was measured on a scale of ≤0.5 cm; (iii) longitudinal follow-up was performed at the same time slot, and in the case of women, the menstrual cycle was considered; (iv) fasting for 2–3 h was required; (v) alcohol intake > 8 h; (vi) urination before the measurement; (vii) restriction of physical exercise for >8 h; (viii) avoidance of medications that affect the water balance measurement (steroids, growth hormone, diuretics); (ix) avoidance of the use of metallic accessories or magnetic targets (rings, bracelets, etc.); (x) the arm was placed at about 30° from the trunk and legs separated by 45°; (xi) the subject was instructed not to move until the weighing was completed; (xii) the subject was placed in the supine position; (xiii) the manufacturer’s protocol was followed; (xiv) the measurement was performed twice and if different results were recorded, a third time was performed.

### 2.3. Second Consultation

In this consultation, the dietary intervention begins with the delivery of the following documents (Appendix A) [8,12,13,37,38,39]:A 7-day menu [38] (pp. 145–147). Details about it are displayed in Section 2.3.1.Recommendations to initiate the method: cooking time of food; quantity of food with the use of food scrapers and ladles to make a single meal in the family home, recommending the weighing of meat, fish, and potatoes; advice on the use of oil or other fats for the consumption of raw or cooked foods; encouraging the consumption of vegetables and fruits; time between meals of around three hours [38] (pp. 148–150).Food equivalents/substitution table, which is a document in which foods are presented in groups that are interchangeable among themselves because they are equivalent in size or weight, in addition, they have a similar nutritional composition.A food survey to record intake, if they were unable to take a meal on any of the days, and the alternative option decided upon [38,39] (p. 137). In this document, the dietary intake was recorded for 7 days of the week and 5 meals of the day (breakfast, morning snack, lunch, afternoon snack, and dinner). It was asked what was consumed and the amount, the type of liquid consumed at these meals (water, soft drinks, juices, alcoholic beverages, etc.), and whether these meals were eaten alone or in company.

In the incorporation of dietary changes, a distinction was made between adults under 65 years of age and older adults (≥65 years). In adults under 65 years of age, a behavioral accompaniment was performed that was differentiated by gender, and the prominence could be shared with the partner or the people with whom he/she lived during the acquisition period (including the daily practice of cooking food); in the case of women, they were usually in charge of food shopping and/or cooking. In the case of older adults (≥65 years), the situation arose that sometimes they were not able to take care of themselves, and it was necessary to count on the collaboration of their descendants or the caregiver in charge of them. Subjects could make enquiries via telephone and/or e-mail. In addition, a monthly meeting of the group was scheduled to discuss topics of interest that would help to better understand the relevance of the nutritional approach at that time of life, or to improve the implementation of complementary recommendations to guarantee adherence to this intervention. These considerations helped to overcome the various barriers to incorporating better dietary habits [8,10,11].

#### 2.3.1. Personalized Diet

The software described by the Professor José Mataix Verdú was used to perform the individual diets: (Version 0698.046, https://www.funiber.org/software-calculo-de-dietas). An energy restriction of about 500–1000 Kcal*day^−1^ was used, according to the recommendations of the Spanish Society for the Study of Obesity and the Academy of Nutrition and Dietetics [13,36] (p. 21, p. 189, p. 167). The FAO/WHO (Food and Agriculture Organization/World Health Organization) formula was used to calculate the recommended energy intake with the current body weight of the subject [44] (p. 167). In summary, the diet delivered was hypocaloric and balanced (45–55% carbohydrate, 15–25% protein, and 25–35% total fat) [36] (p. 190). When possible, drastic changes in energy and macronutrient intakes were avoided compared to pre-intervention intakes. The diet was adapted to taste preferences of the individuals, gradually introducing the consumption of the following groups of food: (i) abundant consumption of fruits and vegetables; (ii) higher consumption of plant-based foods than animal-based foods (legumes, rice, pasta, potatoes, nuts, etc.); (iii) consumption of whole-grain cereals and their derivatives; (iv) regarding to the consumption of foods with higher animal protein content, the following were recommended: dairy products, and their low-fat derivatives (fish, shellfish, crustaceans, poultry, and eggs); (v) limited consumption of red meats, fast assimilation sugars (biscuits, muffins, biscuits, etc.), and beverages with caloric content (orange, lemon, cola, etc.); (vi) it was advised that the use and/or storage of alcoholic beverages (beer, wine, spirits, etc.) should be monitored in the family home; (vii) preferential consumption of olive oil and sunflower oil for raw meals; (viii) limited consumption of added fats in cooking and in the final processing of foods (butter, mayonnaise, sauces, or other similar products). To improve adherence to the intervention, a pattern was initially used in which food was not normally weighed, but the amount of food added on the plate was measured using food ladles and scrapers. The ladles were used for meals, such as lentils, beans, chickpeas, rice stew, soups, among others. The spoons were used for meals, such as dried rice, pasta, etc. For cold meats, such as turkey breast ham, york, or cured serrano ham (or similar), the number of slices and their weight were indicated; in the case of cheeses, the weight or portions were indicated. For meat and fish, the approximate size and/or optimal weight was shown using images. The fruits were consumed by number according to type and for vegetables no maximum limit was indicated [45].

### 2.4. Third and Subsequent Consultations

It was recommended to register a survey of the foods consumed for one or two weeks, if they did not reach the originally planned goal. In addition, they were given additional recommendations on how to deal with meals on weekends, at celebrations away from home, or during holidays. When the change in body composition was not adequate, it was recommended to weigh the food consumed by using a scale and record these details in the dietary survey [38,39] (p. 137).

### 2.5. Statistical Analysis

The Kruskal–Wallis test was used to compare the parameters analyzed before and after the intervention. The odds ratio was calculated through a logistic regression to analyze the relationship of consultation attendance with changes in weight, fat, and fat-free mass. R software was used for the statistical analysis, *p*-values < 0.05 were considered statistically significant [46].

## 3. Results

### 3.1. Baseline Characteristics of the Subjects

A total of 591 subjects participated in this investigation, and 4091 consultations were performed in subjects aged 19 to 86 years. Characteristics of the study subjects were the following: attended for more than 1.5 months (470 vs. 121), females (403 vs. 188), subjects aged 19 and 64 years with obesity (553 vs. 38 and 330 vs. 261), achieved clinically significant changes in fat mass (469 vs. 122), and recommended changes in fat-free mass (530 vs. 61). The characteristics of the subjects showing more adherence by attending more than 1.5 months were: females (326 vs. 144), subjects aged 19 to 64 years old (437 vs. 33), diagnosed with obesity (277 vs. 199), subjects who achieved clinically significant changes in fat mass (413 vs. 57), and recommended changes in fat-free mass (433 vs. 37). Table 1 summarizes the main details of the patients involved in this study: percentage of men and women, age (19–64 years vs. ≥65 years), degree of excess weight (overweight and obesity), different groups of subjects obtaining various types of changes in body weight and fat mass (≥5% change in body weight and fat mass; <5% change in body weight and ≥5% fat mass; ≥5% change in body weight and <5% fat mass; <5% change in body weight and fat mass), and changes in fat-free mass (FFM ≥ 0% vs. FFM < 0%). All these groups were analyzed according to attendance to consultation (≤3 consultations vs. >3 consultations) [35,36,45] (p. 1011). Individuals obtaining changes ≥ 5% fat mass were considered as subjects with clinically significant changes, and those who obtained fat-free mass change ≥ 0% were considered subjects with recommendable changes.

The variables used for the analysis of body composition change were (Table 2, Figure 1a–c): (i) body mass index (BMI, kg/m^2^), weight divided by height^2^; (ii) initial body weight, fat mass, and fat-free mass (BW_i_, FM_i_, FFM_i_) (kg; %); (iii) fat mass index (FMI, kg/m^2^), fat mass divided by height^2^; (iv) fat-free mass index (FFMI, kg/m^2^), fat-free mass divided by height^2^ [30,31,32,45] (p. 381, p. 1011). As a response to the first secondary objective, it is worth mentioning that the comparison of body composition index variables between high and low attendance groups did not reveal statistically significant differences in any of the cases, but it was considered interesting to highlight the differences found in the means and standard deviations, regarding the indexes analyzed (BMI_i_, FMI_i_ y FFMI _i_), 31.75 ± 5.24 kg/m^2^, 11.76 ± 4.03 kg/m^2^, 20 ± 3.06 kg/m^2^) (Table 2 and Figure 1a–c).

### 3.2. Changes in Body Composition

The variables used for the analysis of changes in body composition were (Table 2): (i) BMI changes (BMI_f–i_), final minus initial BMI; (ii) changes in body weight (BW_f–i_), (change in body weight between two consultations × 100) divided by initial body weight; (iii) fat mass change (FM_f–i_), (fat mass change between two consultations × 100) divided by initial fat mass; (iv) fat-free mass change (FFM_f–i_), (fat-free mass change between two queries × 100) divided by the initial fat-free mass; (v) fat mass index changed (FMI_f–i_), final fat mass index minus the initial; (vi) change in fat-free mass index (FFMI_f–i_), final fat-free mass index minus initial. The percentage of changes in body weight and fat mass was calculated based on the initial body weight and fat mass, according to the example indicated by the Endocrine Society Scientific Statement [8] (p. 101). Subjects that attended for more than 1.5 months of consultation obtained greater changes in body composition: BMI_f–I_ (−2.39 ± 1.62 vs. −0.73 ± 0.69 kg/m^2^), BW_f–i_ (−6.45 ± 4.53 vs. −1.93 ± 1.78 kg and −7.46 ± 4.73 vs. −2.35 ± 2.14%), FM_f–i_ (−5.56 ± 4.21 vs. −1.63 ± 1.71 kg and −18.33 ± 13.22 vs. −6.0 ± 6.24%), FMI_f–i_ (−2.05 ± 1.51 vs. −0.61 ± 0.64 kg/m^2^), FFM_f–i_ (−0.89 ± 2.83 vs. −0.3 ± 1.56 kg and 4.1 ± 3.73 vs. 1.2 ± 1.73%), FFMI_f–i_ (−0.33 ± 1.02 vs. −0.12 ± 0.57 kg/m^2^). When comparing the analyzed variables, all cases show significant differences, according to higher or lower consultation attendance; comparisons between BMI_f–i_, BW_f–i,_ FM_f–i_ y FFM_f–i_ revealed differences between the changes presented; however, these differences disappear when comparisons between BMI_f–i_, FMI_f–i_ y FFMI_f–i_ (Table 2 and Figure 1a–c) areas were analyzed.

As a response to the second secondary objective related to body composition variables analyzed, the greatest changes were obtained in subjects who attended the greatest number of consultations, which should be interpreted as positive, except for the non-recommendable change in the fat-free mass variable, which should be considered as negative (Table 2). This information is related to the fact that 79% of the subjects obtained clinically significant changes in fat, and 90% obtained recommendable changes in fat-free mass (Table 1). In relation to the third secondary objective, 11% of the subjects obtained non-recommendable changes in fat-free mass, and their characteristics were: females (49 vs. 12); subjects 19–64 years old (56 vs. 5), overweight (35 vs. 26), and not obtaining clinically significant changes in fat mass (53 vs. 8) (Table 1 and Table 2).

### 3.3. Odds Ratio Analysis

Table 3 indicates the relationship between the number of consultation attendances and changes in body composition parameters by sex, age, BMI, and other factors. This analysis is based on a respective logistic regression analysis, and it reveals that there were associations between consultation attendance among subjects who obtained clinically significant changes in fat mass and recommended changes in fat-free mass versus those who did not. Specifically, results suggest that patients who attended more than three sessions had a 27 times (95% CI = 14.611–52.979) higher likelihood to obtain a change ≥ 5% in body weight and fat mass, than patients that attended three or less sessions. Similarly, patients who attended more than three sessions have a four times (95% CI = 2.287–6.463) higher likelihood to obtain a change > 5% in body weight and ≥5% fat mass than patients that attended three or less sessions. Patients that attended more than three sessions had a 13 times (95% CI = 2.482–254.218) higher likelihood of obtaining a change ≥ 5% in body weight and <5% fat mass, than patients who had three or less sessions. Finally, patients who attended more than three sessions had a three times (95% CI = 1.640–5.03) higher likelihood of obtaining FFM change < 0% than patients who had three or less sessions.

## 4. Discussion

Reducing the prevalence of obesity is a topic of worldwide interest [3,8]. Therefore, the analysis of factors that support changes in body composition becomes important as a potential tool to prevent the development of cardiovascular diseases, type 2 diabetes, musculoskeletal disorders, respiratory diseases, and various types of cancer [1,3]. The options that proved to be most effective included lifestyle modification, diet, and exercise planning [8]. In this context, this study is justified and of interest for daily clinical practice for the following reasons:✓Prevalence of overweight and obesity in Spain, its evolution, and the economic cost in health care [4,5,6] (p. 58).✓In Spain, there is a need to improve eating habits, due to deviation in caloric profile and in macronutrients in adults [7]. The use of personalized diet will help to improve adherence to interventions, and behavioral support will help to remove barriers that limit lifestyle changes in overweight subjects [8,9,10,11].✓The monitoring of body composition changes and realistic planning of body weight changes are an important part of follow-up in an intervention in overweight subjects [13,14,15]. Given that clinical change in fat mass is an underexplored area, the concept of clinically significant body weight change is extrapolated in this study [8,16,17,18,19,20,21,22,23]. In the analysis of fat-free mass changes, its decrease was considered as a non-recommendable change due to contributing to the prognosis of metabolic syndrome [29,30]. The analysis of fat and fat-free mass index provides added information to the fat and fat-free mass data; therefore, they are variables that should be used in interventions of this type [31,32] (p. 381).✓The main differences between the study presented here and those related to it that are previously reported are the following [37,45,47]: the bioimpedance measurement protocol was included; this is the first time that the analysis was carried out according to the groups attending the consultation (Table 1, Table 2 and Table 3); a comparison was made by different age groups and subjects ≤ 25 years were evaluated; two groups of subjects were designed according to the FFM variable (Table 1 and Table 3); to date, no studies of this type have been found in Spain, where the following variables are analyzed (Table 2 and Figure 1a–c): FMI, FFMI, FMI_f–i_, and FFMI_f–i_.

Considering the mentioned aspects, seventy-nine percent of the subjects participating in the study obtained clinically significant changes in fat mass, 90% obtained recommended changes in fat-free mass, and 80% attended the consultation for a period of more than 1.5 months (≥6 weeks). A mean attendance of 8.01 ± 4.31 consultations (≥17 weeks or ≥4 months) was recorded (Table 1 and Table 2). To evaluate the effectiveness in the changes in fat mass, a comparison with the recommendations of some guidelines of changes between 5 and 10% of body weight was performed (American College of Cardiology, American Diabetes Association, American Heart Association, Endocrine Society Scientific Statement, European Association for the Study of Obesity, Obesity Society and World Health Organization) [8,16,17,18,19,20,21,22]. The Diogenes and Look AHEAD studies presented body weight changes between 5 and 8%, and Brown’s review found one study that presented a change ≥ 10% [22,23,48]. In this study, the change in body weight did not reach 10%. However, the change in fat mass did, in subjects who attended consultations for more than 1.5 months (−18.33 ± 13.22 vs. −6.0 ± 6.24%). The changes in fat-free mass were greater in subjects who attended for more than 1.5 months (−0.89 ± 2.83 vs. −0.3 ± 1.56 kg and 4.1 ± 3.73 vs. 1.2 ± 1.73%) (Table 1 and Table 2), but these changes were positive according to the data of the review by Cava et al., in which it is indicated that in some cases, in around 35%, non-recommended changes were recorded [24]. Although changes in fat mass have been associated with metabolic improvements, there are some reservations about its implementation in clinical practice, since the use of BMI is simpler and cheaper [1,2,13,26,27,49]. On the other hand, the use of fat mass and fat-free mass indices seems to be the next step for a more accurate diagnosis than BMI at the beginning of intervention. Perhaps in a more advanced perspective, it would be of interest to make a comparison with the results found in FMI_f–i_ y FFMI_f–i_ (Table 2) [30,31,32]. In summary, in response to the main objective of this study, the personalized diet is shown to be effective when subjects attend the consultation for more than 1.5 months (Table 2). This concept of effectiveness is based on two aspects: the achievement of clinically significant changes in fat mass and the achievement of recommended changes in fat-free mass (Table 1 and Table 3). The use of fat mass and fat-free mass indexes in these interventions seems to be the future in this area of knowledge.

The major limitations in bringing about clinically significant changes in overweight and obesity are the adherence to treatment and the physiological adaptations that promote weight regain [8,9,10,11]. In treatment adherence, personalized diet [9] and behavioral support are shown to be effective tools to overcome barriers in implementing lifestyle changes [8,23,50,51]. Regarding the behavioral support, previous monitoring of favorite places that the person frequently goes to (indoor and outdoor sports venues) could offer a whole picture about daily life habits, so the goal of monitoring could be to re-introduce habits that would improve results in body composition changes and improve adherence to treatment. The adherence to the diet in this study was based on: (i) informing the subject of the health problems they could have by maintaining their excess weight, and the learning process they had to face to achieve changes in body weight, fat mass, and fat-free mass; (ii) making gradual changes in energy and macronutrient intake versus previous intake; (iii) using easily understood complementary recommendations to help the incorporation of these changes in eating habits. In comparison with other studies regarding ton energy restriction, there is agreement in our recommendation ≥Kcal*día^−1^ [8,13,25,36] (p. 21, p. 521, p. 106). Related to macronutrients uses, only the Endocrine Society Scientific establish some recommendations about lipids and proteins, the rest of the studies show differences in recommendations [8,13,25,36,51] (p. 21, p. 521, p. 106). Perhaps with the analysis of fat mass as the effectiveness ib these interventions, a greater consensus can be found in this area. In those subjects who did not obtain the clinically significant changes in fat mass and recommended changes in fat-free mass (122 vs. 469 and 61 vs. 530), the following barriers to introduce changes in their eating habits were observed: difficulties to buy the recommended foods, to prepare the indicated meals, to have at least four meals a day, to stick to the planning of a full day (where the subjects had some meals on one day of the diet and others on another day), to avoid alcohol intake, or to limit meals outside the home. To remove or decrease the percentage of subjects who obtained non-recommendable changes in fat-free mass, the following recommendations are proposed: to increase protein intake (two approaches can be considered: protein uptake between 1.0–1.2 g × kg × day^−1^ or 1.4–2.0 g × kg × day^−1^); cardiorespiratory (“aerobic”) exercise and resistance exercise should be incorporated to maintain muscle mass and increase the muscle strength of this type of patient [13,24,25,52]. The American College of Sports Medicine recommends ≥150 min each week of cardiorespiratory exercise, with a progression of intensity between moderate and vigorous. Based on resistance exercise, each major muscle group should be trained 2–3 times/week, so that the intensity of training will vary depending on the experience and age of the patient [53] (p. 1336). The monitoring of physical exercise and/or daily physical activity with the use of accelerometers will help to monitor the sedentary habits of daily life [54].

Based on the background presented, this study agrees with the perspective of the American Association of Clinical Endocrinologists and American College of Endocrinology, which indicates that new action protocols should be based on adiposity assessment. This new conceptual approach brings a different perspective, focusing the action on the use of BMI [28]. Based on the experience gained from the development of this research, the following recommendations are made for future studies:i.Fat and fat-free mass analysis: their indices will provide greater accuracy for diagnosis in early consultations versus BMI (Figure 1a–c).ii.Monitoring body composition changes: fat mass and fat-free mass should be recorded (Table 2), then some options should be proposed to improve adherence to the diet and obtain clinically significant changes in fat mass and recommended fat-free mass (Table 1 and Table 3):a.Promote attendance to the dietitian’s office for a period of ≥6 weeks, especially in men and those over 65 years of age with overweight and obesity (Table 1).b.Being part of a dynamic learning process divided into several periods will enable the subject to know where he/she currently stands in the intervention (initiation, improvement, and maintenance).c.Use of a hypocaloric and balanced diet: with an energy restriction of 500–1000 Kcal × day^−1^; an intake of 45–55% carbohydrates, 25–35% total fat, and 15–25% protein.d.The complementary recommendations used will allow the subject to know how to incorporate the indicated changes in his/her day-to-day life.

## 5. Conclusions

This quasi-experimental study describes an example of the daily clinical practice of a dietitian (exercising professional activity in the Southeast of Spain, Alicante), in which the effectiveness of changes in body composition is shown as a correlation to the consultation attendance. The effectiveness of changes in body composition is shown as a function of consultation attendance. The information presented and the results derived from this research will help the design of future studies in the following aspects: (i) to optimize the design and implementation of body composition monitoring; (ii) to support the design of future personalized diet studies that achieve greater effectiveness in changes in fat mass and fat-free mass. Thus, this knowledge will contribute to the daily clinical practice of dietitians and of physical activity and public health professionals.

## Figures and Tables

**Figure 1 healthcare-09-01101-f001:**
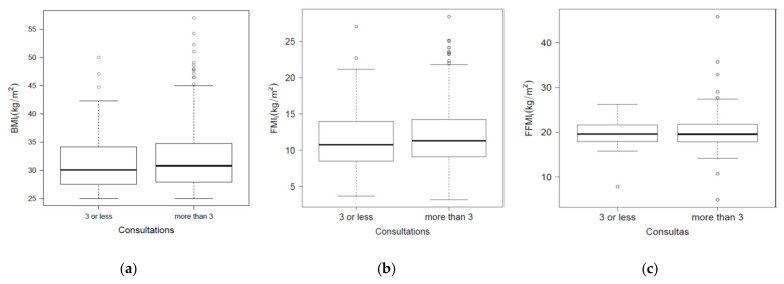
Boxplots comparing the groups based on attendance for: (**a**) BMI_i_; (**b**) FMI_i_ and (**c**) FFMI_i_. body mass index (BMI); fat mass index (FMI); fat-free mass index (FFMI); initial (i); ≤3 consultation attendance (≤1.5 months); attendance to >3 consultations (>1.5 months).

**Table 1 healthcare-09-01101-t001:** Percentage of subjects exposed to dietary intervention (*n*: 591).

Categories	Subcategories	Participants *n* (%)	≤3 Consultation Attendance (*n*: 121)	Attendance to >3 Consultations (*n*: 470)
Sex	Male	188 (32%)	44 (36%)	144 (31%)
	Female	403 (68%)	77 (64%)	326 (69%)
Age	19–64 years old	553 (94%)	116 (96%)	437 (93%)
	≥65 years old	38 (6%)	5 (4%)	33 (7%)
BMI	Obesity	330 (56%)	59 (49%)	272 (58%)
	Overweight	261 (44%)	62 (51%)	199 (42%)
Clinically significant changes	Change ≥ 5% in body weight and fat mass	313 (53%)	15 (12%)	298 (63%)
	Change < 5% in body weight and ≥5% fat mass	156 (26%)	41 (34%)	115 (24%)
Clinically non-significant changes	Change ≥ 5% in body weight and <5% fat mass	11 (2%)	1 (1%)	10 (2%)
	<5% change in body weight and fat mass	111 (19%)	64 (53%)	47 (10%)
Recommended changes	Change in FFM ≥ 0%	530 (90%)	97 (80%)	433 (92%)
Not-recommended changes	FFM change < 0%	61 (10%)	24 (20%)	37 (8%)

*n*: sample; ≤3 consultation attendance (≤1.5 months); attendance to >3 consultations (>1.5 months).

**Table 2 healthcare-09-01101-t002:** Dietary variables based on attendance at the consultation.

Parameters	Min	Med	Max	Mean (SD) (*n*: 591)	≤3 Consultation Attendance (*n*: 121)	Attendance to >3 Consultations (*n*: 470)	KW Statistic (*p*-Value)
Age (years old)	19	42	86	43.43 (13.96)	40.74 (14.38)	44.12 (13.78)	−2.321 (0.021)
BMI_i_ (kg/m^2^)	25	30.67	57.01	31.75 (5.24)	31.28 (5.01)	31.88 (5.29)	−1.166 (0.245)
BW_i_ (kg)	58.6	83.6	166.7	85.38 (15.87)	84.67 (13.82)	85.56 (16.37)	−0.612 (0.541)
FM_i_ (kg)	8.9	29.7	74.8	31.22 (9.98)	30.29 (9.97)	31.46 (9.97)	−1.155 (0.249)
(%)	12.36	37.19	82.93	36.4 (8.28)	35.64 (9.21)	36.59 (8.02)	−1.045 (0.298)
FMI_i_ (kg/m^2^)	3.19	11.13	28.42	11.76 (4.03)	11.4 (4.29)	11.85 (3.95)	−1.046 (0.297)
FFM_i_ (kg)	15.4	50	134	54.16 (11.89)	54.38 (11.18)	54.1 (12.07)	0.239 (0.812)
(%)	17.07	62.81	87.64	63.6 (8.28)	64.36 (9.21)	63.41 (8.02)	1.045 (0.298)
FFMI_i_ (kg/m^2^)	4.86	19.59	45.83	20 (3.06)	19.87 (2.7)	20.03 (3.15)	−0.54 (0.59)
Number of consultations	2	6	42	6.9 (4.42)	2.6 (0.49)	8.01 (4.31)	
BMI_f–i_ (kg/m^2^)	−9.65	−1.72	1.84	−2.05 (1.62)	−0.73 (0.69)	−2.39 (1.62)	16.959 (<0.001)
BW_f–i_ (kg)	−32.4	− 4.6	4.9	−5.53 (4.51)	−1.93 (1.78)	−6.45 (4.53)	17.112 (<0.001)
(%)	−29.81	−5.59	4.91	−6.42 (4.79)	−2.35 (2.14)	−7.46 (4.73)	17.475 (<0.001)
FM_f–i_ (kg)	−25.1	−3.9	1.9	−4.76 (4.15)	−1.63 (1.71)	−5.56 (4.21)	15.809 (<0.001)
(%)	−72.13	−13.04	7.39	−15.8 (13.1)	−6 (6.24)	−18.33 (13.22)	14.806 (<0.001)
FMI_f–i_ (kg/m^2^)	−8.58	−1.46	0.72	−1.75 (1.5)	−0.61 (0.64)	−2.05 (1.51)	15.752 (<0.001)
FFM_f–i_ (kg)	−9.8	−0.8	17.2	−0.77 (2.63)	−0.3 (1.56)	−0.89 (2.83)	3.082 (0.002)
(%)	−3	2.7	23.97	3.5 (3.61)	1.2 (1.73)	4.1 (3.73)	−12.451 (<0.001)
FFMI_f–i_ (kg/m^2^)	−3.96	−0.3	6.47	− 0.29 (0.95)	−0.12 (0.57)	− 0.33 (1.02)	3.074 (0.002)

Standard deviation (SD); Kruskal-–Wallis (KW); maximum (Max); median (Med); minimum (Min); ≤3 consultation attendance (≤1.5 months); attendance to >3 consultations (>1.5 months); end (f); initial (i); body mass index (BMI); body weight changes (BW_f–i_); fat mass changes (FM_i–f_); fat-free mass (FFM); fat mass index (FMI); fat-free mass index (FFMI); fat-free mass changes (FFM_i–f_); fat-free mass changes (FMI_i–f_); fat-free mass index changes (FFM_i–f_).

**Table 3 healthcare-09-01101-t003:** Odds ratio (OR) for the risk of consultations by gender, age, BMI, changes in weight, fat mass, and fat-free mass.

≤3 Consultation Attendance vs. Attend to >3 Consultations				
Categories	Subcategories	OR	95% CI	*p*
Sex	Male vs. female	1.294	0.846–1.960	0.229
Age	19–64 years old vs. ≥65 years old	1.752	0.729–5.205	0.254
BMI	Overweight vs. obesity	1.431	0.959–2.139	0.078
Clinically significant changes vs. Clinically non-significant changes	Change ≥ 5% in body weight and fat mass vs. <5% change in body weight and fat mass	27.052	14.611–52.979	<0.0001
	Change < 5% in body weight and ≥5% fat mass vs. <5% change in body weight and fat mass	3.819	2.287–6.463	<0.0001
Comparison of clinically insignificant changes	Change ≥ 5% in body weight and <5% fat mass vs. <5% change in body weight and fat mass	13.617	2.482–254.218	0.0143
Not recommended changes vs. Recommended changes	FFM change < 0% vs. Change in FFM ≥ 0%	2.896	1.640–5.03	0.0002

≤3 Consultation attendance (≤1.5 months); attendance to >3 consultations (>1.5 months).

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
