# Peer review of "Personalized Diet in Obesity: A Quasi-Experimental Study on Fat Mass and Fat-Free Mass Changes"

_healthcare, 2021, doi:10.3390/healthcare9091101_

Round 1

Reviewer 1 Report

The paper addresses one of the most difficult problems, which is the adherence of obese people to treatment plans. It addresses as it concerns the age groups adults from 19 years to over 65 years. 

The sudy is interesting as it concerns the hypothesis, and has a large amount of subjects thus achieving the necessary statistical significance.

The main finding of the study is that the subjects who followed the consultation intervention, for over 1.5 months achieve better results as it concerns fat, body weight and the rest of the obese related parameters.

The study is interesting in the sense that it provides a minimum time window for consultation intervention to obese subject so that the treatment plan (personalised diet) can be effective.

One of the main issues that could be addressed perhaps as a follow up and perhaps addressed more specifically in the discussion section is the monitoring of the physical activity (PA) on a daily basis, and the monitoring of favorite places that the person is going to (parks, gyms, cafeterias, restaurants etc). In the study grocery stores are referenced, but then, the correlations with the actual BMI evolution is not vlear.

Author Response

Response to Reviewer 1 Comments

The paper addresses one of the most difficult problems, which is the adherence of obese people to treatment plans. It addresses as it concerns the age groups adults from 19 years to over 65 years.

The sudy is interesting as it concerns the hypothesis, and has a large amount of subjects thus achieving the necessary statistical significance.

The main finding of the study is that the subjects who followed the consultation intervention, for over 1.5 months achieve better results as it concerns fat, body weight and the rest of the obese related parameters.

The study is interesting in the sense that it provides a minimum time window for consultation intervention to obese subject so that the treatment plan (personalised diet) can be effective.

Point 1: One of the main issues that could be addressed perhaps as a follow up and perhaps addressed more specifically in the discussion section is the monitoring of the physical activity (PA) on a daily basis, and the monitoring of favorite places that the person is going to (parks, gyms, cafeterias, restaurants etc).

Point 2: In the study grocery stores are referenced, but then, the correlations with the actual BMI evolution is not vlear.

Response to point 1: Thank you very much for this observation. Specific comments related to physical activity have been added in page 12 (line 380). Regarding to monitoring of favourite places, comments have been added in page 12 (lines 357-362).

Response 2: Thank you for your comment. The introduction has addressed some barriers which hinder body composition changes in overweight and obesity, however, lack of time to plan the purchase, preparation and cooking of healthy foods are not factors that have been studied in detail in this research. The objective of the study was to analyse the relationship between the adherence of a personalized diet and the effectiveness of the body composition changes of the patients who participated, paying special interest in fat and fat free mass. We will consider this observation for further studies.

Reviewer 2 Report

The authors present a novel study investigating the impact of a personalized diet and nutrition counseling sessions on body mass composition. In addition, they discuss the benefits of assessing body fat percentage and other parameters in addition to BMI, which has the potential to change our standard of care. 

The manuscript would benefit from a thorough review with a native English speaker, there were many points in the manuscript were the grammar was off and it made it difficult for the reader to ascertain the exact meaning. 

Author Response

Response to Reviewer 2 Comments

Point 1:

The authors present a novel study investigating the impact of a personalized diet and nutrition counseling sessions on body mass composition. In addition, they discuss the benefits of assessing body fat percentage and other parameters in addition to BMI, which has the potential to change our standard of care.

The manuscript would benefit from a thorough review with a native English speaker, there were many points in the manuscript were the grammar was off and it made it difficult for the reader to ascertain the exact meaning.

Response 1: Thank you very much for your nice words about this research. English grammar and spelling have been reviewed through the whole text.